# Evolution of immune responses to SARS-CoV-2 in mild-moderate COVID-19

Adam K. Wheatley [1,2,10], Jennifer A. Juno [1,10], Jing J. Wang[3,10], Kevin J. Selva[1], Arnold Reynaldi [4], Hyon-Xhi Tan[1], Wen Shi Lee [1], Kathleen M. Wragg[1], Hannah G. Kelly[1,2], Robyn Esterbauer [1,2], Samantha K. Davis [1], Helen E. Kent[1,5], Francesca L. Mordant [1], Timothy E. Schlub [4,6], David L. Gordon[7], David S. Khoury [4], Kanta Subbarao [1,8], Deborah Cromer[4], Tom P. Gordon[3,9], Amy W. Chung [1], Miles P. Davenport [4] & Stephen J. Kent [1,2,5✉]

The durability of infection-induced SARS-CoV-2 immunity has major implications for rein-fection and vaccine development. Here, we show a comprehensive profile of antibody, B cell and T cell dynamics over time in a cohort of patients who have recovered from mild-moderate COVID-19. Binding and neutralising antibody responses, together with individual serum clonotypes, decay over the first 4 months post-infection. A similar decline in Spike-specific CD4$^+$ and circulating T follicular helper frequencies occurs. By contrast, S-specific IgG$^+$ memory B cells consistently accumulate over time, eventually comprising a substantial fraction of circulating the memory B cell pool. Modelling of the concomitant immune kinetics predicts maintenance of serological neutralising activity above a titre of 1:40 in 50% of convalescent participants to 74 days, although there is probably additive protection from B cell and T cell immunity. This study indicates that SARS-CoV-2 immunity after infection might be transiently protective at a population level. Therefore, SARS-CoV-2 vaccines might require greater immunogenicity and durability than natural infection to drive long-term protection.

---

[1] Department of Microbiology and Immunology, University of Melbourne, at The Peter Doherty Institute for Infection and Immunity, Melbourne, VIC, Australia. [2] Australian Research Council Centre for Excellence in Convergent Bio-Nano Science and Technology, University of Melbourne, Melbourne, VIC, Australia. [3] Department of Immunology, College of Medicine and Public Health,  Flinders University, Adelaide, SA, Australia. [4] Kirby Institute, University of New South Wales, Kensington, NSW, Australia. [5] Melbourne Sexual Health Centre and Department of Infectious Diseases, Alfred Hospital and Central Clinical School, Monash University, Melbourne, VIC, Australia. [6] Sydney School of Public Health, Faculty of Medicine and Health, University of Sydney, Sydney, NSW, Australia. [7] Department of Microbiology and Infectious Diseases, Flinders University and SA Pathology, Flinders Medical Centre, Adelaide, SA, Australia. [8] WHO Collaborating Centre for Reference and Research on Influenza, The Peter Doherty Institute for Infection and Immunity, Melbourne, VIC, Australia. [9] Department of Immunology, SA Pathology, Flinders Medical Centre, Adelaide, SA, Australia. [10]These authors contributed equally: Adam K. Wheatley, Jennifer A. Juno, Jing J. Wang. ✉email: skent@unimelb.edu.au

Animal studies[1,2] and the scarcity of confirmed re-infection[3] in humans suggests immune protection from SARS-CoV-2 infection is likely, although the durability of this protection is debated. Lasting immunity following acute viral infection often requires maintenance of both serum antibody and antigen-specific memory B and T lymphocytes and is notoriously pathogen specific, ranging from life-long for smallpox or measles[4], to highly transient for common cold coronaviruses (CCC)[5].

Neutralising antibody responses are a likely correlate of protective immunity and exclusively recognise the viral spike (S) protein, predominantly targeting the receptor-binding domain (RBD) within the S1 sub-domain[6]. Multiple reports describe waning of S-specific antibodies in the first 2–3 months following infection[7–12]. However, extrapolation of early linear trends in decay might be overly pessimistic, with several groups reporting that serum neutralisation is stable over time in a proportion of convalescent donors[8,12–17]. SARS-CoV-2-specific B and T cell responses are also readily induced by infection[6,13,18–24], although the longitudinal dynamics of these key memory populations remains poorly resolved.

In this work we quantified a wide range of S-specific antibody and cellular immune responses in serial blood samples over the first 4 months following convalescence from COVID-19.

## Results

**Decay of SARS-CoV-2-specific antibodies over 4 months.** We recruited a longitudinal cohort of 64 participants who recovered from COVID-19 (Supplementary Table 1). A total of 158 samples were collected between day 26 and 149 post-symptom onset, with samples nominally denoted as early (≤50 days), intermediate (50–100 days) and late (≥100 days) convalescence (Fig. 1a). In early convalescence, neutralisation activity was widespread with a median serological titre of 52, which declined to 34 in late convalescence (Fig. 1b). A mixed-effects modelling approach found that a two-phase decay model best fit with the observed decay of neutralisation titres across the cohort ($p < 0.00001$, likelihood ratio test), with rapid decay evident over the first half of our time-series (half-life ($t_{1/2}$) prior to day 70 = 55 days), compared with slower decay in the second half ($t_{1/2}$ from day 70 = 519 days; Fig. 1b). The capacity of immune plasma to inhibit interaction of the SARS-CoV-2 receptor-binding domain (RBD) with soluble hACE2 receptor[19] waned with a similar two-phase decay, dropping more rapidly before day 70 ($t_{1/2} = 238$ days) and slowing after day 70 ($t_{1/2} = 1912$ days; Fig. 1c). Although the decay of neutralisation and RBD-ACE2 binding inhibition responses were similar, subtle differences may relate to neutralising antibody responses directed to responses in the spike protein outside the RBD. The baseline neutralisation response prior to day 50 was, as previously reported[19], higher in participants with moderate-severe COVID-19 (mean titre 1:140) compared to participants with mild COVID-19 (1:92, $p = 0.0505$, two sample $t$-test with equal variance). The decline in the microneutralisation titre was similar across both groups with moderate-severe participants maintaining higher titres at >100 days (1:106 in moderate-severe vs 1:49 in mild disease ($p = 0.0018$, two sample $t$-test with equal variance)).

Plasma antibodies specific for SARS-CoV-2 S antigens (trimeric spike protein (S), S1, S2, and RBD subdomains) and nucleocapsid (N) antigens were quantified longitudinally using a multiplex bead array[25]. In contrast to neutralisation titres, decay of S-specific IgG was best fit by a model of constant decay over the period of observation ($t_{1/2} = 229$ days), with rates of decay divergent for antibodies binding S1 ($t_{1/2} = 115$ days), S2 ($t_{1/2} = 344$ days), and RBD antigens ($t_{1/2} = 126$; Fig. 1d, Supplementary Fig. 1). IgG3

displayed a more rapid, two-phase decline compared to IgG1 (Supplementary Fig. 1). Consistent with a previous report[26], we find N-specific IgG decays significantly more rapidly than S-specific IgG ($t_{1/2} = 71$ and 229 days, respectively, $p < 0.00001$, Fig. 1d). In contrast to IgG, S-specific IgM and IgA1 fit a two-phase decay, with a more rapid early decay ($t_{1/2} = 55$ and 42 days, respectively) followed by a slower decay in late convalescence ($t_{1/2} = 118$ and >1000 days, respectively; Fig. 1d). A comparison of decay rates between neutralising activity and antibody binding demonstrated that early neutralisation decay occurs at a similar rate to the early decline in S, RBD and S1-specific IgM (Fig. 1e, Supplementary Fig. 1). Neutralisation titre at both early and late convalescence was well correlated with serum inhibition of RBD-ACE2 binding and S, S1, and RBD-specific IgG, IgM (and to a lesser extent IgA1) responses, as well as with S2- and N-specific IgG responses (Supplementary Fig. 2). Neutralising activity during early convalescence was the best correlate of long-term maintenance of neutralisation responses (Spearman rho = 0.88, $p < 0.00001$; Supplementary Fig. 2). Serum inhibition of RBD-ACE2 binding inhibition and S1-specific IgG responses in early infection were also well correlated with neutralisation titre in late convalescence (Spearman rho = 0.79, 0.81, respectively; Supplementary Fig. 2). However, in a multiple regression model, once early neutralisation activity was included no other significant predictors were identified ($p > 0.15$ for all other variables).

The decay of polyclonal antibody in plasma may obscure a more complex picture of the dynamics of individual antibody specificities. To resolve longitudinal serological decay at the level of a single clonotype, we adapted a mass spectrometry (MS)-based quantitative proteomics workflow developed for serum autoantibody profiling[27,28] to track unique CDR-H3 peptides matching recovered S-specific immunoglobulins sequences from convalescent participants[19] ($n = 4$; Fig. 2a, b and Supplementary Fig. 3). Consistent with the decay of polyclonal S-specific antibody in the blood, we find a decline in the relative abundance over time for each unique clonotype (Fig. 2c), although absolute rates of decay did vary, suggesting the kinetics might to some degree be clonotype-, epitope-, or subject-specific.

**Spike-specific IgG+ memory B cells expand during convalescence.** Anti-viral memory B and T cell responses will likely make additive contributions to long-term immunological protection against COVID-19. SARS-CoV-2-specific B cell responses were measured longitudinally in 31 participants where sufficient cells were available (Fig. 1a) using flow cytometry and fluorescent S and RBD probes[19]. Following infection, frequencies of IgG+ S-specific memory B cells increased over time irrespective of disease severity (Fig. 3a and b; gating in Supplementary Fig. 4), compared with low background frequencies seen in a cohort of uninfected control participants ($n = 20$; Fig. 3a; Supplementary Table 1). In contrast, S-specific IgA+ MBC frequencies remained relatively stable while IgM+ MBC frequencies decreased (Fig. 3b). IgG+ S-specific MBC remain significantly elevated at the final relative to the first available sampling ($p < 0.0001$, Wilcoxon), contrasting with stable IgA+ ($p = 0.367$, Wilcoxon) and declining IgM+ populations ($p < 0.0001$, Wilcoxon; Fig. 3c). Assessment of the activation status of S-specific IgG+ MBC using CD21/CD27 staining[29] demonstrated decreased proportions of "activated" MBC and a return to a resting (CD27+ CD21+) phenotype over time (Supplementary Fig. 5). Although present at comparatively low frequencies, the dynamics of RBD-specific MBC largely mirrored that of the parental S-specific population (Supplementary Fig. 6). Modelling the growth rates reveals IgG+ S-specific MBC frequencies had a doubling time of 48 days in early convalescence, after which point the doubling time slowed

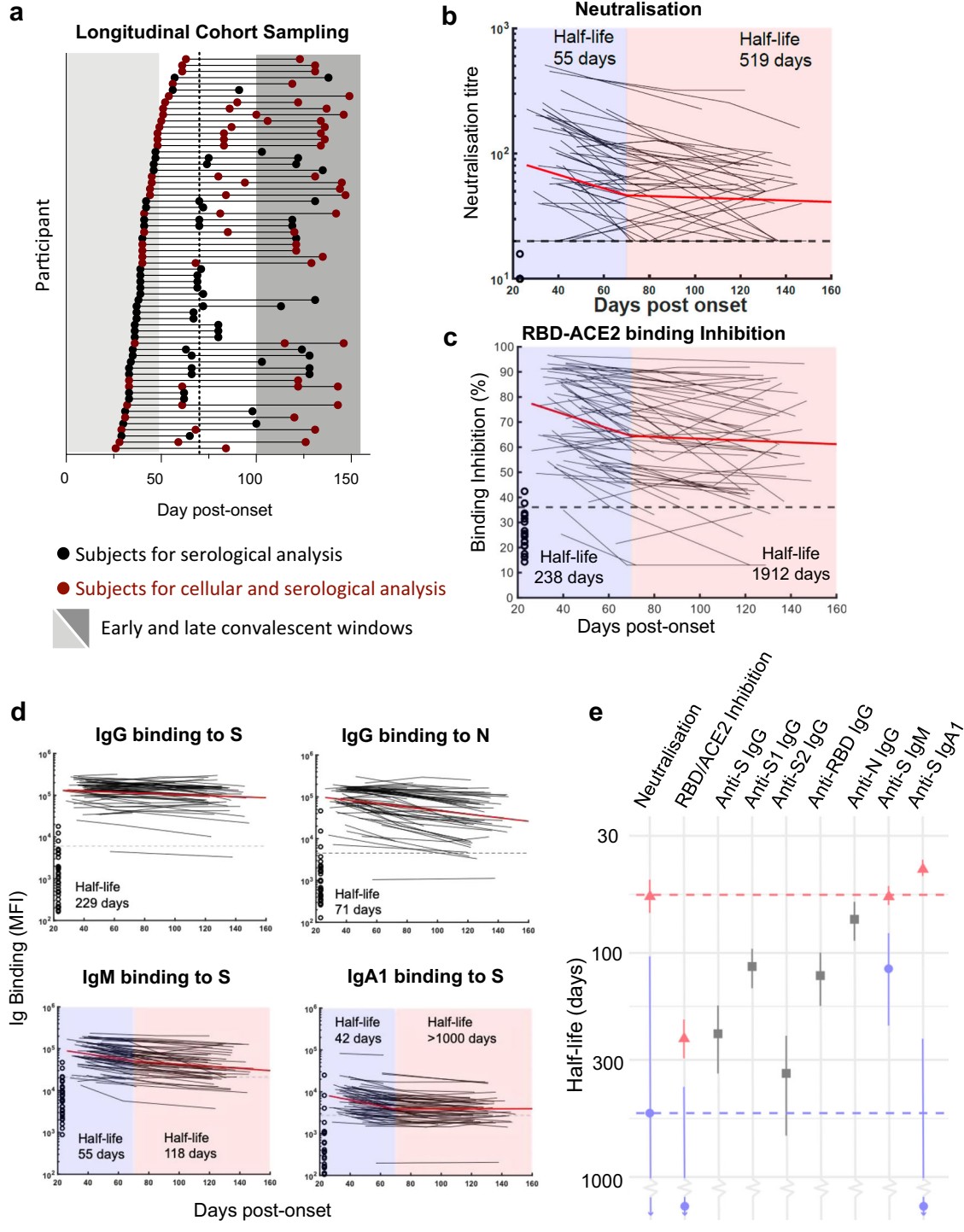

**Fig. 1 Dynamics of serological responses to SARS-CoV-2. a** Timeline of sample collection for each cohort participant ($n = 64$ participants, 158 total samples). Samples included only in serological analysis are indicated in black ($n = 33$); samples included in both serological and cellular immune analysis are indicated in red ($n = 31$). Shaded areas indicate early (<50 days) and late (>100 days) convalescent time periods, and dashed line indicates day 70 midpoint. **b** Longitudinal microneutralisation endpoint titre and **c** inhibition of ACE2 binding (%) for individuals. Best fit two-phase decay slope (red line) is indicated. Uninfected control participants ($n = 26$ for ACE2 binding inhibition and $n = 7$ for microneutralisation) are shown on the left side of each graph. Horizontal dashed lines indicate the upper 90th percentile value of the uninfected control cohort for RBD-ACE2 inhibition and a conservative threshold of 1:20 for microneutralisation. **d** Individual kinetics and best fit decay slopes for IgG binding to spike (S), IgG binding to nucleoprotein (N), IgM binding to S and IgA1 binding to S. $N = 63$ for IgA1. Uninfected control participants ($n = 32$) are shown on the left side of each graph and horizontal dashed lines indicate the upper 90th percentile value of the uninfected control cohort. **e** Estimated half-life and confidence intervals of the neutralising antibody titre before day 70 (red) and after day 70 post-symptom onset (blue) are indicated as dashed vertical lines. Estimated early decay rates and confidence intervals for serological inhibition of ACE2 and antibody binding titres are indicated (single phase decay is shown in grey, two-phase decay indicated in red/ blue). Horizontal dashed lines indicate the median value of the uninfected control cohort ($n = 32$). If no dashed line is shown, the control cohort median lies at or below the *y*-axis limit. Source data are provided as a Source Data file.

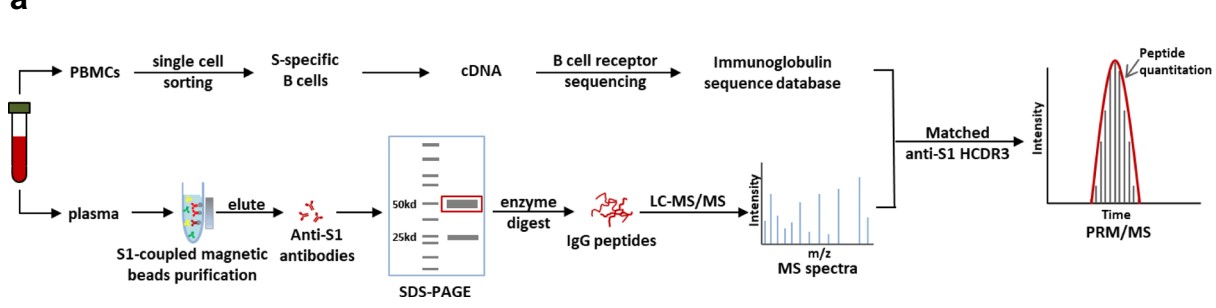

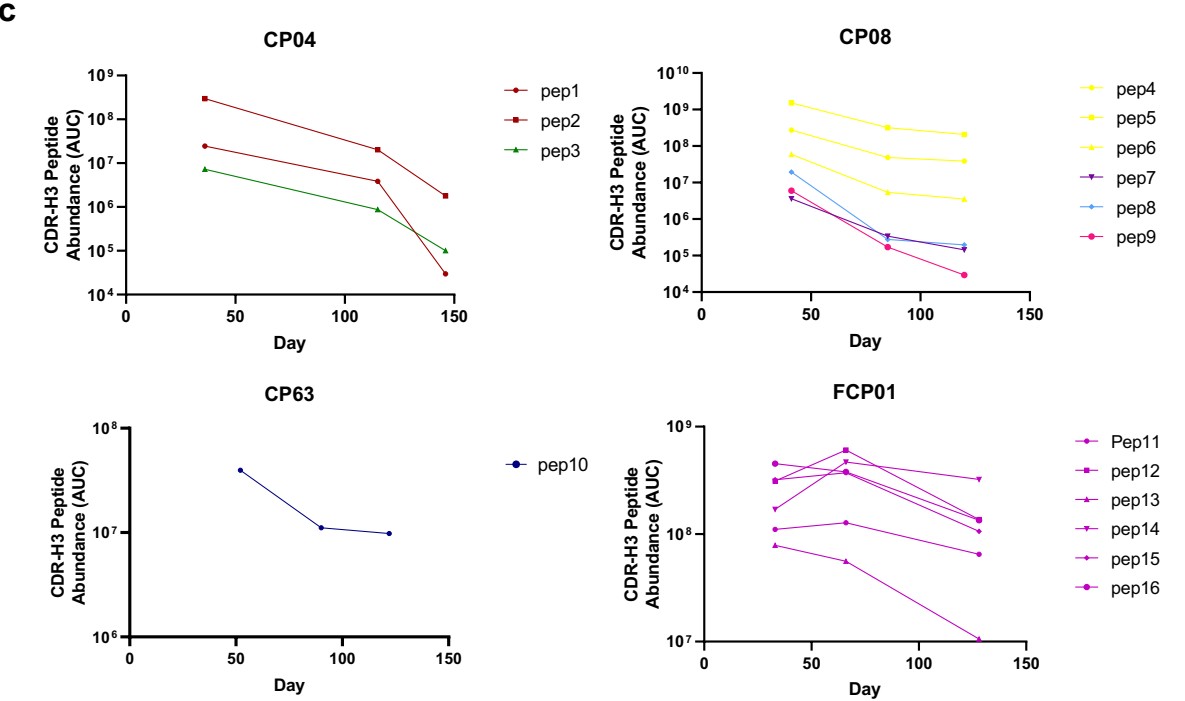

to 843 days, with IgG+ RBD-specific cells MBC broadly comparable (doubling time = 58 days early, $t_{1/2}$ = 247 days late; Supplementary Fig. 7). The consistent and sustained increase in S-specific IgG+ MBC frequencies over time aligns with a prior report of SARS-CoV-2 convalescent participants[13], and with reports from other viral infections[30] or replicating viral vaccines[30–32]. Given the relatively low level of somatic mutation observed in S-specific antibodies recovered from convalescent participants to date[6,19] comparative studies at 6-9 months post-infection will be informative to understand the maturation of the humoral response over time and the protective potential of stably retained MBC populations.

**Fig. 2 Mass spectrometry-based quantification of immunoprecipitated S1-specific clonotypic antibodies. a** Combined B cell receptor sequencing and proteomics platform enables identification and quantification of circulating anti-S1 antibodies. S1-specific IgG was purified from plasma of SARS-CoV-2 convalescent participants using antigen-coupled magnetic beads and heavy chains subject to LC-MC/MS. Peptide spectra are searched against B cell receptor sequencers recovered from single sorted S-specific memory B cells from the same individuals to identify clonotypes based upon CDR-H3 amino acid sequence. Clonotype specific peptides are then used as barcodes for relative quantitative parallel reaction monitoring (PRM) for tracking in longitudinal plasma samples. Targeted peptides are monitored during elution from HPLC and individual peptides quantified based on abundance chromatography curves. PBMCs peripheral blood mononuclear cells, LC-MS/MS liquid chromatography mass spectrometry/mass spectrometry, CDR-H3 heavy-chain third complementarity-determining region. **b** Clonotypes identified based on matched CDR-H3 sequences from S1-specific plasma IgG and B cell receptor sequences from SARS-CoV-2 convalescent participants ($n = 4$). **c** Longitudinal changes in the relative plasma abundance of anti-S1 clonotypes within four convalescent participants over time. The quantity of each reference peptide is expressed as area under the curve (AUC) derived from extracted ion chromatography. For participants CP04, CP08, and CP63, each experiment was repeated independently three times with similar results and data are presented as mean value. For subject FCP01, experiment was repeated independently twice with similar results, and data are presented as the mean value. Source data are provided as a Source Data file.

**Decay in spike-specific T cell responses over 4 months**. Antiviral memory T cell responses have been associated with amelioration of disease for respiratory infection such as influenza[33]. S-specific cTFH and conventional CD4+ and CD8+ memory T cells (Tmem), were quantified using activation induced marker (AIM) assays[19,22] (gating shown in Supplementary Fig. 8) following stimulation with overlapping S (split into S1 or S2) peptide pools (Fig. 4a–c). Responses among the convalescent cohort were substantially more robust than those observed among most uninfected participants (Fig. 4a–c). Frequencies of S-specific memory T cells among convalescent individuals were dynamic over time and varied between participants (Fig. 4d). Pairwise comparison of CD4+ Tmem or cTFH frequencies at the final visit relative to the first available sampling demonstrated a significant reduction in S-specific responses over time ($p = 0.0224$ for CD4+ Tmem, $p = 0.0031$ for cTFH, Wilcoxon), although some participants demonstrated substantial decay and others maintained more stable levels (Fig. 4e). Nonetheless, S-specific CD4+ Tmem responses remained significantly elevated compared to uninfected controls (UI) at both the first and last visits (median 0.077% UI, 0.718% first visit and 0.436% final visit; $p < 0.0001$ for UI vs first visit; $p = 0.0003$ for UI vs final visit, Kruskal–Wallis test). In contrast, frequencies of S-specific CD8+ Tmem were stable at a population level ($p = 0.3247$, Wilcoxon), although individual responses were varied (Fig. 4d). Similar to the CD4+ T cell responses, S-specific CD8+ Tmem responses were significantly higher among convalescent donors compared to controls at both the first and final visits (median 0.1% for UI, 0.312% for first visit, 0.284 for final visit; $p = 0.0057$ for UI vs first visit; $p = 0.0264$ for UI vs final visit, Kruskal–Wallis test). Modelling of the decay rates estimated $t_{1/2}$ of 128 days for cTFH (95% CI 67, 1247) and 119 days for CD4+ Tmem (95% CI 66, 612; Supplementary Fig. 9). In contrast, the estimated decay of CD8+ responses is not significantly different from 0 ($t_{1/2} = 670$ days, 95% CI 97, −136; Supplementary Fig. 9). Therefore, while we and others[22,34] find that CD4+ responses are generally higher during early convalescence, CD8+ T cell responses appear relatively stable during late convalescence.

Multiple studies have reported the presence of CCC cross-reactive CD4+ T cells in a proportion of SARS-CoV-2 uninfected participants[22,34,35], which is consistent with the observed responses in our uninfected controls (Fig. 4a, e). To understand whether such responses might the influence the decay of T cells responses directed toward either S1 or S2 epitopes, we contrasted S1 and S2 responses among the CD4+ T cell subsets. For cTFH, a significant drop in S1 responses was observed over time ($p = 0.0028$, Wilcoxon), while S2 responses were comparably stable but did similarly trend downward ($p = 0.0657$, Wilcoxon) (Supplementary Fig. 10a, b). Analogous patterns were observed for the CD4+ Tmem cells (Supplementary Fig. 10c, d).

Consequently, S2-specific cTFH and CD4+ Tmem populations predominated over S1-directed responses ($p = 0.0147$ and $p = 0.0021$, respectively, Wilcoxon) in late convalescence (Supplementary Fig. 10b, d), possibly reflecting maintenance of cross-reactive responses to CCC. Although a previous report has suggested an increased level of cross-reactivity among C-terminal peptide pools compared to N-terminal peptide pools in uninfected donors[23], we did not find any significant enrichment of cross-reactive CD4 T cell responses among the S2 peptide pool compared to the S1 pool (not shown).

Polyclonal T cell responses to S comprise an array of immunodominant and subdominant epitopes; we therefore additionally tracked single CD4+ T cell epitopes in a subset of 9 donors (Supplementary Fig. 11a). Strikingly, we observed substantial inter- and intra-individual variability in longitudinal epitope-specific responses (Supplementary Fig. 11b, c); in some participants, all epitope-specific responses tracked similarly while in others distinct epitope-specific responses would vary independently over time. In most, but not all, cases, peptide responses tracked similarly between the cTFH and Tmem populations (Supplementary Fig. 11c). Overall, some degree of T cell immunity remains readily detectable in most participants 4 months after infection, although longitudinal epitope-specific frequencies were markedly less predictable.

**Modelling the decay in neutralising antibody responses**. Deconvoluting the protective potential of the suite of concomitant immune responses elicited by SARS-CoV-2 infection is challenging. The general decline of serological immunity over time (Fig. 5a) was similarly observed for most memory immune cell subsets except for IgG+ and IgA+ MBC populations (Fig. 5b). Importantly, rates of immune decay are likely to stabilise over time to levels of homeostatic maintenance[36], although this set point is not yet clear for SARS-CoV-2. Neutralising antibody is the most widely accepted protective correlate against a range of human respiratory viruses[37]. However, any relationship between in vitro neutralisation titres and in vivo protection for SARS-CoV-2 is unclear at present. We therefore developed a simulation model (see Methods) employing the estimated initial distributions of neutralisation titres and decay rates across participants, to predict the time for titres to drop below a nominated cut-off of 1:40, selected based on the 1:40 hemagglutination inhibition titre (a surrogate for neutralisation activity) widely used as the 50% protective titre for influenza[38]. Notably, 43% of our cohort were already below this threshold in early convalescence, with 64% of participants dropping below this threshold in late convalescence. Simulating a population of 1000 individuals, and running the model 1000 times, we find the median time for 50% of the population to drop below a titre of 1:40 was 74 days (Fig. 5c; 95% confidence interval 46 to

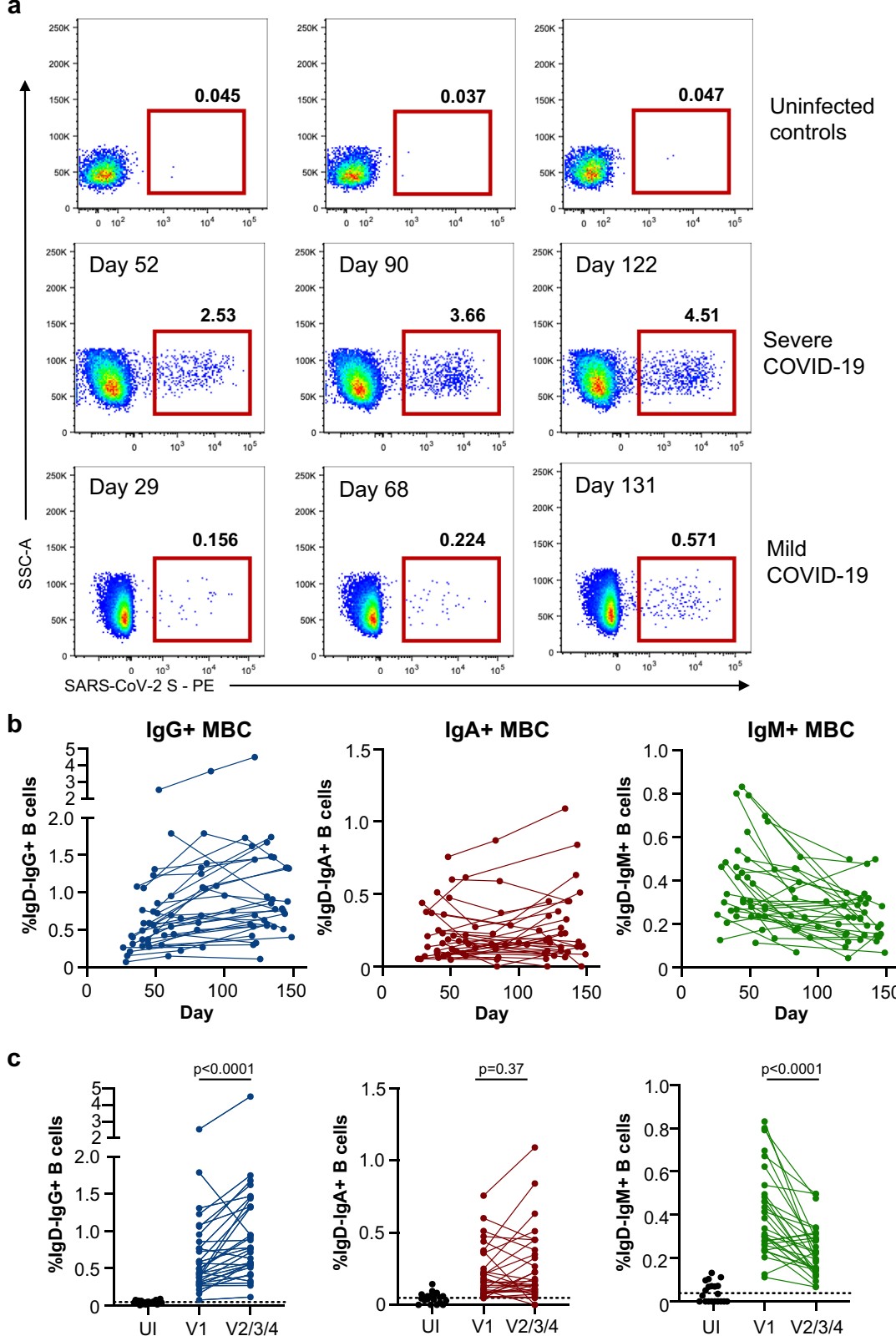

**Fig. 3 Quantification of S-specific memory B cell responses. a** Staining class-switched B cells (CD19 + IgD-) with SARS-CoV-2 spike probes allows the tracking of antigen-specific cells in participants previously infected with SARS-CoV-2, shown relative to uninfected controls. Longitudinal plots from a single individual with severe infection or a single individual with mild infection are shown. **b** Frequencies of S-specific IgG+, IgA+, or IgM+ memory B cells as a proportion of CD19+ CD20+ IgD− B cells in PBMC samples were assessed longitudinally (*n* = 31 participants). **c** Comparison of S-specific IgG+, IgA+, or IgM+ memory B cell frequencies at the earliest and latest timepoint available for each individual (*n* = 31). Median background in uninfected (UI) controls (*n* = 20) is indicated. Statistics assessed by two-tailed Wilcoxon test. Source data are provided as a Source Data file.

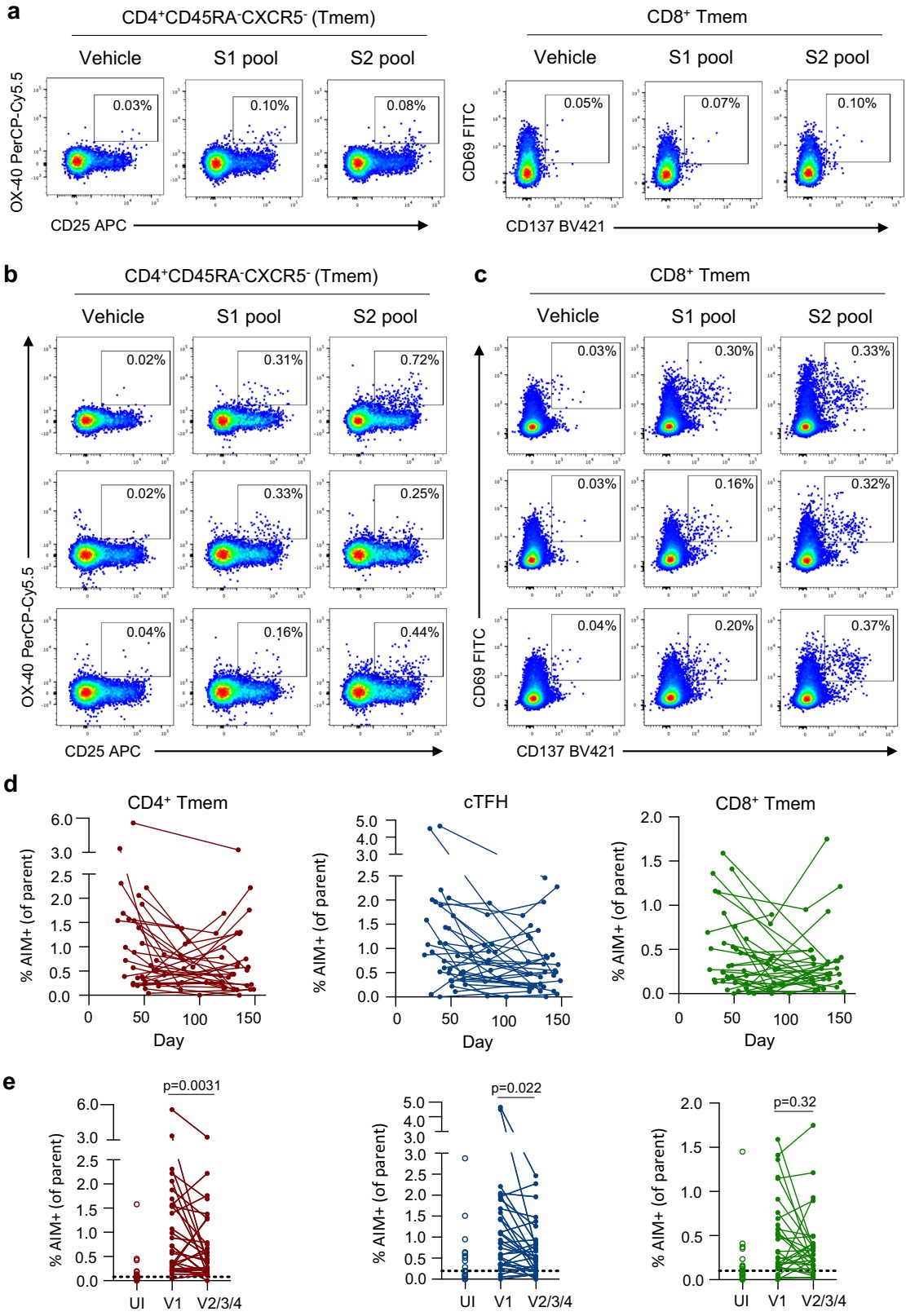

>1000 days). Assuming early neutralisation titres predicted titres into late convalescence, our simulation also allows us to estimate how higher initial levels of neutralisation may affect the proportion of individuals maintaining titres above 1:40. We found that if aiming for a median of 50% of individuals with a titre above 1:40 at 1 year, initial neutralisation titres at about day 30 would need to be in the order of 2.1-fold higher than that observed in our convalescent cohort (95% CI = no increase to 16.9-fold increase required). It is important to emphasise that at present the in vitro neutralisation titre required and the additive contribution of other immune responses to protective immunity are unknown. Similar modelling using a cut-off neutralising

**Fig. 4 Quantification of antigen-specific CD4+ and CD8+ T cell responses. a** Representative staining of AIM markers (CD25, OX-40) on CD4+ Tmem cells (CD3+ CD4+ CD8-CD45RA-CXCR5-) or AIM markers (CD69, CD137) on CD8+ Tmem cells (CD3+ CD8+ CD4-non-naïve) after stimulation with vehicle, S1 or S2 peptide pools in an uninfected individual. **b** Representative staining of AIM markers on CD4+ Tmem cells in longitudinal samples from 1 participant (top row, day 33; middle row, day 61; bottom row, day 143). **c** Representative staining of AIM markers on CD8+ Tmem cells in longitudinal samples from 1 participant (top row, day 41; middle row, day 85; bottom row, day 120). **d** Longitudinal changes in the frequency of total S (S1 + S2 pool responses after background subtraction)-specific responses among CD4+ Tmem, cTFH, and CD8+ Tmem subsets (n = 31). **e** Comparison of S-specific T cell responses at the earliest and latest timepoint available for each individual (n = 31). Statistics assessed by two-tailed Wilcoxon test. Source data are provided as a Source Data file.

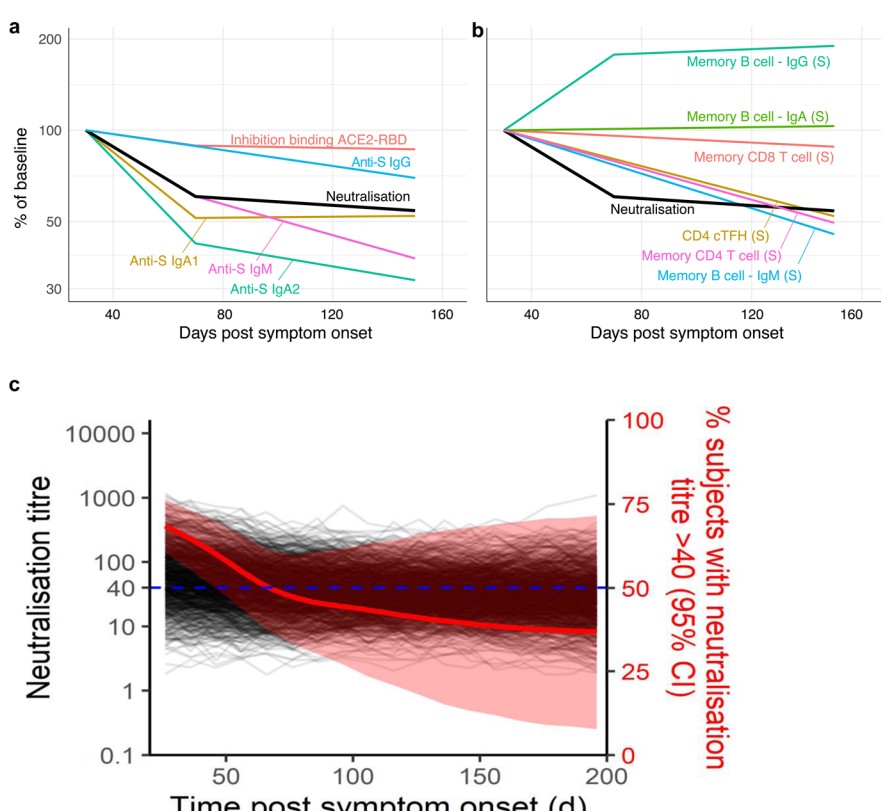

**Fig. 5 Modelling of concomitant immune responses after COVID-19. a** Rates of decay of serological neutralisation activity, ACE2 binding inhibition, and S-specific IgG, IgM and IgA following recovery from SAR-CoV-2 infection. Baseline levels are set to 100% of that of the mean responses at visit 1 (median 40 days, shown in Fig. 1a). **b** Fitted Growth and decay rates for S-specific memory T cell and B cell frequencies in PBMC. The neutralising antibody decay curve (black line) is shown for comparison purposes. **c** Simulation of elicitation and decay of serological neutralisation activity in 1000 individuals based on distributions observed in our SARS-CoV-2 convalescent cohort. The simulation was repeated 1000 times to estimate the proportion of individuals maintaining a neutralisation titre above 1:40 across multiple simulations (median and 95% confidence intervals shown in red). Source data are provided as a Source Data file.

antibody titre of 1:20 suggests that the median time for 50% of the population to drop below a titre of 1:20 is 341 days and that the neutralising titre at day 30 would only need to be 1.07 fold higher than that observed in our cohort to achieve 50% of individuals remaining above a titre of 1:20 by one year (Supplementary Fig. 12). In addition, our analysis assumes that immunity to vaccination decays at a similar rate to infection, and that the decay of neutralisation titre from day 70 to around 140 predicts immune decay over the first year. Despite the limitations inherent in these assumptions, this analysis provides an approach to estimating the target level of immune response necessary for effective vaccination.

## Discussion
We found that both neutralising and binding antibody responses decay after recovery from COVID-19, assessed using both

polyclonal assays and at the level of single antibody clonotypes with a mass spectrometry-based assay. While incredibly durable protective antibody responses have been reported for other viral infections such as measles and smallpox[4], our data suggests that SARS-CoV-2 may mirror immunity to endemic CCC, where serum antibody responses decline and susceptibility to homologous virus re-infection occurs within 1–2 years[5]. A protective threshold for neutralising antibody responses to SARS-CoV-2 has not been defined to date and may be lower than levels of 1:40 or 1:20 we modelled.

Neutralising antibody is a presumed but not yet proven correlate of immune protection for SARS-CoV-2. We observed a more rapid decay of the neutralising response over the first 70 days post-infection ($t_{1/2}$ 55 days) which then slowed ($t_{1/2}$ 519 days). The rapid early neutralisation decay corresponded to the loss of S-specific IgM and recent IgM depletion experiments show IgM contributes to the early neutralisation response[39].

Assuming similar immune kinetics, our modelling suggests SARS-CoV-2 vaccines would likely need to elicit substantially more potent neutralising titres than infection to induce durable protection. Encouragingly, some early vaccine candidates have exceeded this metric when compared against sera from convalescent participants in clinical trials reported to date[40,41].

Persistence of serum antibody is unlikely to be the sole determinant of long-lasting immunity, with anamnestic recall of stably maintained memory T and B cell populations likely reducing infection or disease. The magnitude, quality and protective potential of cellular responses against SARS-CoV-2 requires further definition. T cell memory in the blood contracts several months post-infection although responses were remarkably variable, likely in part reflecting HLA-restricted responses to specific epitopes as we illustrated across 7 CD4 T cell epitopes followed over time (Supplementary Fig. 10). Further studies will be required to more specifically dissect changes in epitope immunodominance over time during convalescence. The consistent rise in S-specific IgG+ memory B cells to a median level of ~0.8% of all IgG+ memory B cells by 4 months suggests even mild-moderate COVID-19 induces substantial cellular immune memory. We speculate that cellular immune memory may reduce rates of re-infection.

## Methods

**Ethics statement**. The study protocols were approved by the University of Melbourne Human Research Ethics Committee (#2056689) and the Southern Adelaide Clinical Human Research Ethics Committee (#39.034), and all associated procedures were carried out in accordance with the approved guidelines. All participants provided written informed consent in accordance with the Declaration of Helsinki.

**Participant recruitment and sample collection**. Participants who had recovered from COVID-19 were recruited through contacts with the investigators and invited to provide serial blood samples. We recruited participants who were convalescent from COVID-19 (>28 days post initial symptoms) who had either a prior +ve nasal PCR during early infection for SARS-CoV-2 or clear exposure to SARS-CoV-2 and a positive ELISA for SARS-CoV-2 S and RBD protein[19]. Contemporaneous controls who did not experience any symptoms of COVID-19 and who were confirmed to be seronegative were also recruited. Participants characteristics of SARS-CoV-2 convalescent and control participants are collated in Supplementary Table 1. For all participants, whole blood was collected with sodium heparin anticoagulant. Plasma was collected and stored at −80 °C, and PBMCs were isolated via Ficoll-Paque separation, cryopreserved in 10% DMSO/FCS and stored in liquid nitrogen.

**Microneutralisation assay**. SARS-CoV-2 isolate CoV/Australia/VIC01/2020[42] was passaged in Vero cells and stored at −80 °C. Plasma was heat-inactivated at 56 °C for 30 min. Plasma was serially-diluted 1:20 to 1:10240 before addition of 100 TCID$_{50}$ of SARS-CoV-2 in MEM/0.5% BSA and incubation at room temperature for 1 h. Residual virus infectivity in the plasma/virus mixtures was assessed in quadruplicate wells of Vero cells incubated in serum-free media containing 1 μg/ml TPCK trypsin at 37 °C/5% CO2; viral cytopathic effect was read on day 5. The neutralising antibody titre is calculated using the Reed/Muench method[43,44]. All samples were assessed in two independent microneutralisation assays.

**Expression of SARS-CoV-2 proteins**. A set of proteins was generated for serological and flow cytometric assays. The ectodomain of SARS-CoV-2 (isolate WHU1;residues 1–1208) was synthesised with furin cleavage site removed and P986/987 stabilisation mutations[19], a C-terminal T4 trimerisation domain, Avitag and His-tag, expressed in Expi293 cells and purified by Ni-NTA affinity and size-exclusion chromatography using a Superose 6 16/70 column (GE Healthcare). SARS-CoV S was biotinylated using Bir-A (Avidity). The SARS-CoV-2 RBD[45] with a C-terminal His-tag (residues 319-541; kindly provided by Florian Krammer) was similarly expressed and purified.

**SARS-CoV-2 bead-based multiplex assay**. The isotypes and subclasses of SARS-CoV-2-specific antibodies were analysed[25]. Briefly, a panel of SARS-CoV-2 antigens including trimeric S, S1 (Sino Biological), S2 (ACROBiosystems), NP (ACROBiosystems,), and RBD[46] were coupled to magnetic COOH- bioplex beads (Biorad) using a two-step carbodiimide coupling reaction. Twenty microlitre of bead mixture containing 1000 beads per region and 20 μl of 1:200 diluted plasma were added per well. SARS-CoV-2- specific antibodies were detected using phycoerythrin (PE)-conjugated mouse anti-human pan-IgG, IgG1, IgG2, IgG3, IgA1,

or IgA2 (Southern Biotech) at 1.3 μg/ml, 25 μl per well. For the detection of IgM, biotinylated mouse anti-human IgM (mAb MT22; MabTech) was added at 1.3 μg/ml, 25 μl per well followed by streptavidin-PE (SA-PE; Thermo Fisher) at 1 μg/ml. Plates were acquired by a FLEXMAP 3D (Luminex). Median fluorescence intensity (MFI) for each isotype/subclass detector was assessed. Background subtraction was conducted, removing background of blank (buffer only) wells. Multiplex assays were repeated twice as two independent experiments.

**RBD-ACE2 binding inhibition multiplex bead-based assay**. RBD protein was coupled to bioplex beads (Biorad) as described above. 20 μl of RBD multiplex bead suspension containing 500 beads per well, 20μl of biotinylated Avitag-ACE2 (kindly provided by Dale Godfrey and Nicholas Gherardin), final concentration of 12.5 μg/ml per well, along with 1:100 dilution of each subject's plasma were added to 384-well plates. Plates were covered and incubated at room temperature (RT) while shaking for 2 h, and then washed twice with PBS containing 0.05% Tween20 (PBST). Biotinylated Avitag-ACE2 was detected using 40 μl per well of SA-PE at 4 μg/ml, incubated with shaking for 1 h at RT. Ten microlitre of PE-Biotin amplifier (Thermo Fisher) at 10 μg/ml was added and incubated for 1 h with shaking at RT. Plates were washed and acquired on a FLEXMAP 3D (Luminex). Anti-SARS-CoV-2 RBD neutralising human IgG1 antibody (SAD-S35, ACROBiosystems, USA) was included as a positive control, in addition to COVID-19 negative plasma and buffer only negative controls. The MFI of bound ACE2 was measured after background subtraction of no ACE2 controls. Maximal ACE2 binding MFI was determined by buffer only controls. % ACE2 binding inhibition was calculated as 100% − (% ACE2 binding MFI per sample/Maximal ACE2 binding). RBD-ACE2 binding inhibition multiplex assays were repeated independently twice.

**Flow cytometric detection of S-specific and RBD-specific memory B cells**. Probes for delineating SARS-CoV-2 S-specific B cells within cryopreserved human PBMC were generated by sequential addition of streptavidin-PE (Thermofisher) to trimeric S protein biotinylated using recombinant Bir-A (Avidity). SARS-CoV-2 RBD protein was directly labelled to APC using an APC Conjugation Lightning-link kit (Abcam). Cells were stained with Aqua viability dye (Thermofisher). Monoclonal antibodies for surface staining included: CD19-ECD (J3-119) (Beckman Coulter), CD20 Alexa700 (2H7), IgM-BUV395 (G20-127), CD21-BUV737 (B-ly4), IgD-Cy7PE (IA6-2), IgG-BV786 (G18-145) (BD), CD14-BV510 (M5E2), CD3-BV510 (OKT3), CD8a-BV510 (RPA-T8), CD16-BV510 (3G8), CD10-BV510 (HI10a), CD27-BV605 (O323) (Biolegend), IgA-Vio450 (clone) (Miltenyi). Cells were washed, fixed with 1% formaldehyde (Polysciences) and acquired on a BD LSR Fortessa or BD Aria II. Gating is shown in Supplementary Fig. 4.

**Quantification of S-specific cTFH, memory CD4+, and memory CD8+ T cells**. Cryopreserved human PBMC were thawed and rested for 4 h at 37 °C. Cells were cultured in 96-well plates at $1 \times 10^6$ cells/well and stimulated for 20 h with 2 μg/peptide/mL of peptide pools (15mer, overlapping by 11) covering the S1 or S2 domains of SARS-CoV-2. Selected donors were also stimulated with SEB (1 μg/mL) as a positive control, or individual peptides at 2 ug/mL: NCTFEYVSQPFLMDL (S1 epitope; previously described in ref. [45]); LPIGINITRFQTLLA (S1 epitope); GWTFGAGAALQIPFA (S2 epitope); ALQIPFAMQMAYRFN (S2 epitope); LLQYGSFCTQLNRAL (S2 epitope;[19,45]); QALNTLVKQLSSNFG (S2 epitope). Following stimulation, cells were washed, stained with Live/dead Blue viability dye (ThermoFisher), and a cocktail of monoclonal antibodies: CD27 BUV737 (L128), CD45RA PeCy7 (HI100), CD20 BUV805 (2H7), (BD Biosciences), CD3-BV510 (SK7), CD4 BV605 (RPA-T4), CD8 BV650 (RPA-T8), CD25 APC (BC96), OX-40 PerCP-Cy5.5 (ACT35), CD69 FITC (FN50), CD137 BV421 (4B4-1) (Biolegend), and CXCR5 PE (MU5UBEE, ThermoFisher). Cells were washed, fixed with 1% formaldehyde and acquired on a BD LSR Fortessa using BD FACS Diva. Gating is shown in Supplementary Fig. 7.

**MS-based quantitative proteomics of serum anti-S1 antibodies**. The workflow for anti-S1 proteomic profiling is shown in Fig. 2a. Briefly, antibodies against SARS-CoV-2 S1 spike protein were affinity-purified from convalescent plasma of COVID-19 participants at different time points using S1 protein-coupled magnetic beads (Acrobiosystems). IgG heavy chains were isolated after reduced SDS-PAGE and digested with trypsin and chymotrypsin to generate peptides for LC-MS/MS using a Thermo Scientific Orbitrap Exploris 480 mass spectrometer coupled to an Ultimate 3000 UHPLC (Dionex). De novo sequencing data analysis was performed by Peaks studio X-plus software (Bioinformatics Solution). Peptide sequences were referenced against recovered heavy-chain immunoglobulin sequences generated from single sorted S-specific memory B cells[19] to identify matched CDR-H3 peptides. Anti-S1 clonotypic antibody expression levels were monitored by parallel reaction monitoring (PRM) as described previously[28,47]. Fragment ion extracted ion chromatograms (XICs) per CDR-H3 peptide were visualized in Skyline version 20.1.0.155 (University of Washington) and inspected manually to ensure correct assignments. The annotated spectra of individual peptides and their corresponding XICs are shown in Supplementary Figure 3.

**Estimating the decay rates**. We sought to predict the response variable ($y_{ij}$ for patient i at timepoint j) as a function of days post-symptom onset, assay replicate

(as a binary categorical variable) and a random effect for each individual (both in intercept and slope). The dependency of the response variables on days post-symptom onset can be modelled by using one slope decay (as in Eq. (1) below) or two slope decay (as in Eq. (2), using $s_{ij}$ from Eq. (3) below). The model can be written as below:

$$y_{ij} = \beta_0 + b_{0i} + \beta_1 R_{ij} + \beta_2 t_{ij} + b_{2i} t_{ij} - \text{for a model with a single slope; and} \quad (1)$$

$$y_{ij} = \beta_0 + b_{0i} + \beta_1 R_{ij} + \beta_2 t_{ij} + b_{2i} t_{ij} + \beta_3 s_{ij} + b_{3i} s_{ij} - \text{for a model with two}$$
$$\text{different slopes, in which :} \quad (2)$$

$$s_{ij} = \begin{cases} 0, & t_{ij} < T_0 \\ t_{ij} - T_0, & t_{ij} \geq T_0. \end{cases} \quad (3)$$

The parameter $\beta_0$ is a constant (intercept), and $b_{0i}$ is a patient-specific adjustment to the overall intercept. The slope parameter $\beta_2$ is a fixed effect to capture the decay slope before $T_0$; which also has a subject-specific random effect $b_{2i}$. To fit a model with two different decay rates, an extra parameter $\beta_3$ (with a subject-specific random effect $b_{3i}$) was added to represent the difference between the two slopes. Assay variability between replicates was modelled as a single fixed effect $\beta_1$, in which we coded the replicate as a binary categorical variable $R_{ij}$. We calculated the Akaike Information Criterion values for each different model with different random effect structures and found that the inclusion of of all random effects is supported in most variables.

The response variables obtained were highly variable, containing zeros where the value was below the limit of detection and contrasted with samples where very high levels were observed. Thus, we performed log transformations of the non-zero data to help normalize variability and censored every value less than 40 for the microneutralisation data; every value less than 0.01 for the T cell and B cell data; and every negative value for the multiplex data. More specifically, a mixed-effect regression method that allows for censoring at the limit of detection was used to estimate the parameters in the model. This was done by using *lmec* version 1.0 library in *R*, using the ML algorithm to fit for the fixed effects. We also tested if the decay of serological response variables was fitted better by a single or two different decay slopes (likelihood ratio test—based on the likelihood value and the difference in the number of parameters). 95% CI for the fixed effect parameters was calculated based on the standard error estimates, which can be obtained directly by using the *varFix* function from *lmec* library. These analyses were carried out in *R* version 4.0.2.

**Simulating the decay of serological neutralisation activity**. To understand the decay in serum neutralisation we employed a simulation approach using the parameters estimated from our mixed-effect censoring regression model of decay. The fixed effect estimates averaged the intercept across experimental replicates ($\beta_0 + \beta_{1/2}$ from equations above) and random effects were randomly selected from a multivariate normal distribution with covariance matrix taken from the mixed-effect regression with censoring lmec object. The residual error standard deviation for simulated data every 10 days was taken from the lmec object. The confidence interval for the percentage of participants with a neutralisation titre above 1:40 was estimated empirically with the percentile method by repeating the simulation 1000 times, where for each replicate the fixed effects were drawn from a normal distribution based on their standard error (as well as randomly selected random effects). To estimate the fold increase in initial neutralisation titre required to achieve >50% of individuals with a titre above 40 at 1 year we assumed that the rate of decay was constant from day 70 onwards and projected forward the expected titres in the simulated populations. The median and confidence intervals for the proportion of individuals with titre >40 were calculated from these 1000 simulated populations.

**Statistical analyses**. Associations between neutralisation, inhibition of ACE2 binding and antibody binding were assessed using both Spearman correlation, and multiple regression (R version 4.0.2). The geometric mean of replicate neutralisation measurements and the arithmetic mean of replicate measurements in other assays were used in the correlation and regression analyses for other measurements. Neutralisation titres below the limit of detection (a titre of 20) were assigned the arbitrary value 10 prior to calculating the geometric mean for the purposes of the Spearman correlation, where rank and not magnitude of the measurements is important. For the multiple regression analysis values below the limit of detection were set at the detection threshold and censoring regression was performed using the function *censReg* (from the *censReg* library version 0.5-30) to determine which measurements during early convalescence were significant predictors of neutralisation titre during late convalescence. Comparison of B and T cell frequencies at first and final sampling was performed using Wilcoxon Rank Sum test in GraphPad Prism 8. Unpaired analysis of antigen-specific T cell frequencies between uninfected controls and convalescent donors at first or final visits was performed using a Kruskal–Wallis test with Dunn's multiple comparisons post-test comparing both convalescent groups to the uninfected controls. All statistical tests used were two sided.

**Reporting summary**. Further information on research design is available in the Nature Research Reporting Summary linked to this article.

## Data availability

All data are provided in the article and Supplementary Information files or from the corresponding author upon reasonable request. Source data are provided with this paper.

## Code availability

All data analysis code are available from the corresponding author upon reasonable request.

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

## Acknowledgements

We thank the participations for the generous involvement and provision of samples. We thank E. Haycroft, E. Lopez, C. Nelson and T. Amarasena and C. Batten (University of Melbourne) for excellent technical assistance. The SARS-CoV-2 RBD expression plasmids were kindly provided by F. Krammer (Icahn School of Medicine at Mt Sinai). Recombinant human ACE2 was kindly provided by N. Gherardin and D. Godfrey (University of Melbourne). We acknowledge the Melbourne Cytometry Platform (Melbourne Brain Centre node) for provision of flow cytometry services. We thank T. Chataway and A. Colella (Flinders Proteomics Facility) for technical support with quantitative proteomics. This study was supported by the Victorian Government, an Australian government Medical Research Future Fund award GNT2002073 (S.J.K., M.P.D. and A.K.W.), the ARC Centre of Excellence in Convergent Bio-Nano Science and Technology (S.J.K.), an NHMRC program grant APP1149990 (S.J.K. and M.P.D.), NHMRC project grant GNT1162760 (A.K.W.), an NHMRC-EU collaborative award APP1115828 (S.J.K. and M.P.D.), the European Union Horizon 2020 Research and Innovation Programme under grant agreement 681137 (S.J.K.), Emergent Ventures Fast Grants (A.W.C.), the Jack Ma Foundation (K.S.), and the A2 Milk Company (K.S.). J.A.J., D.S.K. and S.J.K. are supported by NHMRC fellowships. J.J.W. is supported by Flinders University DVCR Fellowship and Flinders Health & Medical Research Institute COVID-19 Research Grant. A.K.W., K.S., D.C. and M.P.D. are supported by NHMRC Investigator grants. The Melbourne WHO Collaborating Centre for Reference and Research on Influenza is supported by the Australian Government Department of Health.

## Author contributions

A.K.W., J.A.J., J.J.W., H.-X.T., T.E.S., D.L.G., D.S.K., D.C., T.P.G., A.W.C., M.P.D. and S.J.K. designed the study and experiments. A.K.W., J.A.J., J.J.W., K.J.S., A.R., H.-X.T., W.S.L., K.M.W., H.G.K., R.E., S.K.D., H.E.K., F.L.M., T.E.S. and K.S. performed experiments. A.K.W., J.A.J., J.J.W., H.-X.T., W.S.L., A.R., D.S.K., T.E.S., D.C., K.S., M.P.D. and S.J.K. analysed the experimental data. A.K.W., J.A.J., J.J.W., K.J.S., T.E.S., D.S.K., D.C., M.P.D. and S.J.K. wrote the manuscript. All authors reviewed the manuscript.

## Competing interests

The authors declare no competing interests.
