## [Peer Review File · Nature Communications]

REVIEWER COMMENTS

Reviewer #1 (Remarks to the Author):

The authors have adequately addressed my concerns.

Reviewer #2 (Remarks to the Author):

The authors have thoroughly gone through my comments and answered all questions. I have nothing more to add.

Reviewer #3 (Remarks to the Author):

The authors have addressed most of my previous concerns with the notable exception of the placement of dashed lines indicative of where uninfected controls score on particular assays. As mentioned before, this is meant to demonstrate the specificity of their assays. Although the authors have now provided dashed lines indicating the median, this is not appropriate for these studies. The median shows where 50% of the negative controls score – in other words this is the cutoff value for 50% assay specificity. I should have been clearer in my prior request – the authors should provide the 32 individual data points from the naïve controls in the figures (this was done for fig. 3 and fig. 4 but is still missing from Fig. 1) and add the dashed lines at the point in which there is 90% specificity (not 50% specificity as the dashed lines are currently drawn). This may be critical to the interpretation of the antibody half-lives if the 50% of naïve control samples that we can't see here turn out to score appreciably high. However, it is not possible to determine if this is an issue until the authors provide the individual data points from the naïves in Fig. 1 in the same way that they provided this information in the other figures. If a sizeable number of the COVID lines fall below the dashed line of 90% specificity, then the authors may need to recalculate antibody half-lives based on samples that score above this threshold in addition to their current analysis based on the low-specificity thresholds.

Reviewer #3:

The authors have addressed most of my previous concerns with the notable exception of the placement of dashed lines indicative of where uninfected controls score on particular assays. As mentioned before, this is meant to demonstrate the specificity of their assays. Although the authors have now provided dashed lines indicating the median, this is not appropriate for these studies. The median shows where 50% of the negative controls score – in other words this is the cutoff value for 50% assay specificity. I should have been clearer in my prior request – the authors should provide the 32 individual data points from the naïve controls in the figures (this was done for fig. 3 and fig. 4 but is still missing from Fig. 1) and add the dashed lines at the point in which there is 90% specificity (not 50% specificity as the dashed lines are currently drawn). This may be critical to the interpretation of the antibody half-lives if the 50% of naïve control samples that we can't see here turn out to score appreciably high. However, it is not possible to determine if this is an issue until the authors provide the individual data points from the naïves in Fig. 1 in the same way that they provided this information in the other figures. If a sizeable number of the COVID lines fall below the dashed line of 90% specificity, then the authors may need to recalculate antibody half-lives based on samples that score above this threshold in addition to their current analysis based on the low-specificity thresholds.

Response

We now illustrate the individual background data points from the 32 uninfected control samples on the left of each antibody profile panel and add a dashed line for the 90% specificity for each of the antibody profiles in Figure 1D and Supplementary Figure 2 as suggested. With a 90% threshold, we assessed whether a sizeable proportion (defined as >25%) of the antibody responses from the first sampling fall below this high 90% threshold, which may reduce the accuracy of the calculated decay half-lives as the reviewer suggests. Using these criteria, the antibody profiles of IgG2 to all antigens and IgM, IgA1 and IgA2 to N, IgM to RBD and IgA2 to S, have antibody levels where a sizeable proportion of the antibody responses from the first sampling falls below this high 90% threshold of the uninfected controls. Removing the low values only and calculating the decay from the higher antibody samples would provide a skewed view of decay of these immune responses. We have therefore elected to remove the half-life calculations of this subset of antibody responses measured. The other antibody responses have no or only a limited number of data points from the first time point that approach a 90% background specificity threshold and will have more accurate calculations of half-lives.

The revised Figure 1D and Supplementary Figure 2 are shown below. We have revised the Figure 1 and Supplementary Figure 2 legends to note the addition of the control data and the change in calculation of the dotted line background and also noted in Figure S2 the antibody responses above which are closer to levels in uninfected controls that we have removed the half-life calculation.

Figure 1D

Figure 1D Legend:

Dynamics of serological responses to SARS-CoV-2

(D) Individual kinetics and best fit decay slopes for IgG binding to spike (S), IgG binding to nucleoprotein (N), IgM binding to S and IgA1 binding to S. N=63 for IgA1. Uninfected control participants (n=32) are shown on the left side of each graph and horizontal dashed lines indicate the upper 90th percentile value of the uninfected control cohort.

Supplementary Figure 2

Supplementary Figure 2 legend:

Fitting of the decline in antibody binding across different immunoglobulin isotypes.

The best-fit model and half-lives are shown for the fitting of the decay of antibody binding to different SARS-CoV-2 antigens (n=64 subjects). Two-phase decay is indicated by red (before day 70) and blue (after day 70) shaded areas. No shading indicates where single-phase decay provided the best fit.

Uninfected control participants (n=32) are shown on the left side of each graph and horizontal dashed lines indicate the 90th percentile value of the uninfected control cohort. Note that for IgG2 to all antigens, IgM, IgA1 and IgA2 to N, IgM to RBD, and IgA2 to S, the SARS-CoV-2 infected cohort has a sizeable proportion (>25%) of responses at the first time point that are below the 90th percentile of the 32 uninfected controls and we have not calculated decay half-lives for these responses.

REVIEWERS' COMMENTS

Reviewer #3 (Remarks to the Author):

The authors have now added dashed lines indicating the 90% naïve control threshold to Figure 1D. However, we still do not have naïve control samples provided in Fig 1B and Fig 1C. If the naïve controls are below detection ($<1:10$ for Fig. 1B or below 5% binding inhibition for Fig. 1C) then the number of naïve control samples tested should be mentioned in the results and/or figure legend along with the description of these results. If one or more of the naïve controls scores in the detectable range ($>1:10$ or $>5\%$), then all of the naïve controls should be added to the graph and a dashed line provided at the 90% specificity threshold, similar to the data provided in the revised Fig 1D.

This question was raised in the first round of review:

"The authors should include at least 20 naïve control serum samples in their neut assays, RBD/ACE inhibition assays.... "

Reviewer #3 (Remarks to the Author):

The authors have now added dashed lines indicating the 90% naïve control threshold to Figure 1D. However, we still do not have naïve control samples provided in Fig 1B and Fig 1C. If the naïve controls are below detection (<1:10 for Fig. 1B or below 5% binding inhibition for Fig. 1C) then the number of naïve control samples tested should be mentioned in the results and/or figure legend along with the description of these results. If one or more of the naïve controls scores in the detectable range (>1:10 or >5%), then all of the naïve controls should be added to the graph and a dashed line provided at the 90% specificity threshold, similar to the data provided in the revised Fig 1D.

Authors Reply:

We now provide the naïve uninfected control data on figures 1B and 1C. We provide a dashed line at the 90% specificity threshold as requested for RBD-ACE2 binding inhibition and have revised the legend accordingly. As advised in the response to the previous review for the microneutralisation assay, negative control samples are routinely assayed as are control samples obtained from the NIBSC. These samples are reliably negative, giving no neutralising signal at <1:20 dilutions. We now provide additional data on plasma from 7 uninfected controls that were run contemporaneously in the microneutralization assay with the samples from the COVID-19 cohort. These samples were run with a lower initial dilution than the convalescent samples (1:10, versus 1:20 respectively), however 6/7 controls had neutralisation titres of <1:10, with a single subject having some neutralisation at 1:10 but none at 1:20 (with the estimated 90% specific threshold being ~1:15). We therefore conservatively indicate 1:20 as a reliable cut-off for neutralising signal in Fig 1B. The revised portion of the legend and the Figure 1 are shown below.

Figure 1 Legend (B) Longitudinal microneutralisation endpoint titre and **(C)** inhibition of ACE2 binding (%) for individuals. Best fit two-phase decay slope (red line) is indicated. Uninfected control participants (n=26 for ACE2 binding inhibition and n=7 for microneutralisation) are shown on the left side of each graph. Horizontal dashed lines indicate the upper 90th percentile value of the uninfected control cohort for RBD-ACE2 inhibition and a conservative threshold of 1:20 for microneutralisation.